# The rs8176740 *T/A* and rs512770 *T/C* Genetic Variants of the *ABO* Gene Increased the Risk of COVID-19, as well as the Plasma Concentration Platelets

**DOI:** 10.3390/biom12040486

**Published:** 2022-03-23

**Authors:** Gilberto Vargas-Alarcón, Julian Ramírez-Bello, Rosalinda Posadas-Sánchez, Gustavo Rojas-Velasco, Alberto López-Reyes, Laura Martínez-Gómez, Silvestre Ortega-Peña, Isela Montúfar-Robles, Rosa Elda Barbosa-Cobos, Marva Arellano-González, José Manuel Fragoso

**Affiliations:** 1Department of Molecular Biology, Instituto Nacional de Cardiología Ignacio Chávez, Mexico City 14080, Mexico; gvargas63@yahoo.com (G.V.-A.); marva_01@hotmail.com (M.A.-G.); 2Department of Endocrinology, Instituto Nacional de Cardiología Ignacio Chávez, Mexico City 14080, Mexico; dr.julian.ramirez.hjm@gmail.com (J.R.-B.); rossy_posadas_s@yahoo.it (R.P.-S.); 3Intensive Care Unit, Instituto Nacional de Cardiología Ignacio Chávez, Mexico City 14080, Mexico; gustavorojas08@gmail.com; 4Laboratory of Gerosciences, Instituto Nacional de Rehabilitación “Luis Guillermo Ibarra Ibarra”, Mexico City 14289, Mexico; allorey@yahoo.com (A.L.-R.); laurae.mtzg@gmail.com (L.M.-G.); silvestreortega@yahoo.com.mx (S.O.-P.); 5Research Unit, Hospital Juárez de México, Mexico City 07760, Mexico; ismontufar@gmail.com; 6Rheumatology Service, Hospital Juárez de México, Mexico City 07760, Mexico; rebcob@yahoo.com

**Keywords:** COVID-19, genetics, human ABO blood group system, SARS-CoV-2, susceptibility

## Abstract

We conducted a case-control study in order to evaluate whether *ABO* gene polymorphisms were associated with a high risk of developing COVID-19 in a cohort of patients. Six *ABO* gene polymorphisms (rs651007 *T*/*C*, rs579459 *T*/*C*, rs495828 *T/G*, rs8176746 *A/C*, rs8176740 *T/A*, and rs512770 *T*/*C*) were determined using TaqMan genotyping assays in a group of 415 COVID-19 patients and 288 healthy controls. The distribution of rs651007 *T*/*C*, rs579459 *T*/*C*, rs495828 *T*/*G*, and rs8176746 *A*/*C* polymorphisms was similar in patients and healthy controls. Nonetheless, under co-dominant (OR = 1.89, pC_Co-dominant_ = 6 × 10^−6^), recessive (OR = 1.98, pC_Recessive_ = 1 × 10^−4^), and additive (OR = 1.36, pC_Additive_ = 3 × 10^−3^) models, the *TT* genotype of the rs8176740 *T/A* polymorphism increased the risk of developing COVID-19. In the same way, under co-dominant, recessive, and additive models, the *TT* genotype of the rs512770 *T*/*C* polymorphism was associated with a high risk of developing COVID-19 (OR = 1.87, pC_Co-dominant_ = 2 × 10^−3^; OR = 1.87, pC_Recessive_ = 5 × 10^−4^; and OR = 1.35, pC_Additive_ = 4 × 10^−3^, respectively). On the other hand, the *GTC* and *GAT* haplotypes were associated with a high risk of COVID-19 (OR = 5.45, pC = 1 × 10^−6^ and OR = 6.33, pC = 1 × 10^−6^, respectively). In addition, the rs8176740 *TT* genotype was associated with high-platelet plasma concentrations in patients with COVID-19. Our data suggested that the *ABO* rs512770 *T/C* and rs8176740 *T/A* polymorphisms increased the risk of developing COVID-19 and the plasma concentration of platelets.

## 1. Introduction

The severe acute respiratory syndrome coronavirus 2 (SARS-CoV-2) produced coronavirus disease 2019 (COVID-19), a pandemic that has resulted in 416,614,051 confirmed cases globally, with 5,844,097 deaths as of 17 February 2022 [1,2,3]. SARS-CoV-2 virus is of the order Nidovirales and belongs to the coronaviridae family [4]. SARS-CoV-2 is composed of a ribonucleic acid (RNA) genome that contains in itself a nucleocapsid (N) in a helical shape, covered by an envelope lipid that contains three main proteins, the envelope protein (E), the spike protein (S), and the membrane protein (M). The E and M proteins are responsible for the assembly of the virus to host, and S protein is responsible for the entry of the virus [5,6,7]. The S protein is a homotrimer that is composed of two subunits, the union-receptor S1 subunit (subdivided into the S1A domain or N-terminal domain and the S1B domain or receptor-binding domain) and the S2 subunit of the membrane fusion that facilities the entry of viral genome [5,6,7,8].

Recent evidence has showed that the SARS-CoV-2 virus infects the host cell through S1 protein union with the angiotensin-2 receptor (ACE-2) [7,8,9,10]. Nonetheless, recent studies have shown that some molecules, such as the CD-147 receptor, dipeptidyl peptidase 4 (DPP4) receptor, and transmembrane protease serine type 2 (TMPRSS2), as well as the ABO blood groups, could be involved in this phenomenon [11,12,13,14,15]. The ABO blood groups (A, B, AB, and O) are a non-modifiable inherited trait, which plays an important role in the first defense against bacterial and viral infections [14,15]. Recent experimental studies have found that the ABO blood groups played an important role as an inhibitor of the adhesion of the viral S protein to an ACE-2 expressing cell line, and revealed that these groups may prevent the attachment of the viral S protein and the ACE-2 receptor [14,15,16,17,18]. In addition, recent studies have reported that the A blood type A is associated with a high probability of being infected with the SARS-CoV-1 and SARS-CoV-2 viruses compared to the O blood type, which has a significantly low risk of infection. This suggested that blood type O may be a protective factor against infection [14,19,20,21,22,23].

The blood group ABO gene, located in the q34.2 region of chromosome 9, codes for a glycosyltransferase that adds the ultimate monosaccharide to a glycoconjugate and forms the A or B blood group’s specific antigen. Recent studies have associated some single nucleotide polymorphisms (SNPs) of this gene with cardiovascular diseases, hypertension, and SARS-CoV-2 infection [10,14,24,25,26,27].

In this context, considering the critical role of the ABO blood groups as the first line of defense against infection and entry of the virus into the host, the present study aimed to establish whether the rs651007 *T/C,* rs579459 *T*/*C*, rs495828 *T/G*, rs8176746 *A*/*C*, rs8176740 *T*/*A*, and rs512770 *T*/*C* polymorphisms increased the risk of developing COVID-19. In addition, we evaluated whether these polymorphisms were associated with biochemical markers of damage by the SARS-CoV-2 virus in a sample of COVID-19 patients.

## 2. Materials and Methods

### 2.1. Characteristics of the Study Population

This study included 703 Mexican mestizos, 415 patients with SARS-CoV-2 infection confirmed by RT-PCR test in at least one biological sample (61% men and 39% women, with a mean age of 52.2 ± 14.8 years) and 288 healthy controls. The patients were recruited from April 2020 to February 2021 from different hospital centers: 162 from the Instituto Nacional de Rehabilitación “Luis Guillermo Ibarra”, 108 from the Hospital Juárez de México, and 145 from the Instituto Nacional de Cardiología Ignacio Chávez. The diagnosis of COVID-19 was made based on clinical characteristics such as loss of taste, odor, dry cough, fatigue, fever, diarrhea, chills, nasal congestion, sore throat, conjunctivitis, headache, musculoskeletal pain, skin rashes, dizziness, heart rate, and oxygen saturation, together with a positive test for SARS-CoV-2 [1,2,3]. The control group included 288 healthy individuals (60% men and 40% women, with a mean age of 33.1 ± 7.6 years) that worked in the intensive care unit attending patients with COVID-19. This group control included medical residents, laboratory personnel, and nurses. In addition, all controls submitted to a SARS-CoV-2 antibody test (Elecsys^®^ Anti-SARS-CoV-2, Roche Diagnostics International Ltd. CH-6343, Rotkreuz, Switzerland) in order to rule out infection by SARS-CoV-2. The study complied with the Declaration of Helsinki and was approved by the Ethics and Research Commission of Instituto Nacional de Cardiología Ignacio Chávez (project number 21-1237). All patients or their relatives signed the institutional consent letter.

### 2.2. Laboratory Analyses

The blood samples were treated and processed in the laboratory’s class II biological safety cabinet following the institutional security protocols, and the guidelines established in the Official Mexican Standards NOR-007-SSA3-2011, NOM-087-SEMARNAT-SSA1-2002, NOM-010-SSA2-2010, and NMX-EC-15189 IMNC-2015.

In addition, the biomarkers of damage by the SARS-CoV-2 virus in the kidney, lung, and liver were defined as follows: creatinine > 1.3 mg/dL, ferritin > 300 ng/dL, lactic acid dehydrogenase (LDH) > 160 U/L, C reactive protein (CRP) >10 mg/dL, total bilirubin > 1.2 mg/dL, aminotransferase alanine (ALT) > 35 U/L, aminotransferase aspartate (AST) > 35 U/L, hemoglobin > 17 g/dL, and platelets > 350 × 10^3^/μL according to the guidelines establish in the MSD Manual Professional Version 2018 (https://www.msdmanuals.com/professional/SearchResults?query=Normal+laboratory+values (accessed on 6 June 2021)). On the other hand, according to the guidelines established in the MSD Manual, type 2 diabetes mellitus (T2DM) was defined as a fasting glucose ≥110 mg/dL and hypertension with a systolic blood pressure ≥140 mmHg and/or diastolic blood pressure ≥90 mmHg (https://www.msdmanuals.com/professional/SearchResults?query=Normal+laboratory+values (accessed on 6 June 2021)).

### 2.3. Genetic Analysis

DNA extraction was performed from peripheral blood according to the method of Lahiri and Nurnberger [28].

Following the manufacturer’s instructions, the *ABO* 5′UTR (rs651007 *T/C*, rs579459 *T/C,* and rs495828 *T/G*)*, ABO* Leu266Met rs8176746 *A/C*, *ABO* Phe216Ile rs8176740 *T/A*, and *ABO* Ser74Pro rs512770 *T/C* polymorphisms were genotyped using TaqMan genotyping assays on a 7900HT Fast Real-Time PCR system (Applied Biosystems, Foster City, CA, USA). Ten percent of the samples were determined twice to avoid genotyping errors, and all results were concordant.

### 2.4. Statistical Analysis

The analysis of data was performed with SPSS version 18.0 (SPSS, Chicago, IL, USA). Either the Mann–Whitney U test or Student’s t-test were used for the comparison of continuous variables in COVID-19 patients. In addition, for categorical variables, Chi2 or Fisher’s exact tests were performed. The allele and genotype frequencies of *ABO* polymorphisms on patients with COVID-19 and healthy controls were obtained by direct counting. The Chi2 test evaluated the Hardy–Weinberg equilibrium. The associations of the polymorphisms with COVID-19 were evaluated by logistic regression analysis under the following inheritance model: additive (major allele homozygotes versus heterozygotes versus minor allele homozygotes), codominant (major allele homozygotes versus minor allele homozygotes), dominant (major allele homozygotes versus heterozygotes + minor allele homozygotes), over-dominant (heterozygotes versus major allele homozygotes + minor allele homozygotes), and recessive (major allele homozygotes + heterozygotes versus minor allele homozygotes), adjusting for age and gender. All *p*-values were corrected (pC) by the Bonferroni test. The values of pC < 0.05 were considered statistically significant, and all odds ratios (OR) were presented with 95% confidence intervals. In this study, COVID-19 occurrence was based in the following OR cases: (a) OR = 1 did not affect the odds of developing COVID-19, (b) OR > 1 was associated with higher odds of developing COVID-19, and (c) OR < 1 was associated with lower odds of developing COVID-19. The haplotype construction and linkage disequilibrium analysis (LD, D’) were performed using Haploview version 4.1 (Broad Institute of Massachusetts Institute of Technology and Harvard University, Cambridge, MA, USA). The biochemical markers in the COVID-19 patients were compared in the different genotypes. The data were expressed as means ± SD, and comparisons were performed by ANOVA and least significant difference (LSD) as post hoc test; *p* values < 0.05 were considered statistically significant. The statistical power to detect an association with COVID-19 was 0.80 according to the OpenEpi software (http://www.openepi.com/SampleSize/SSCC.html (accessed on 17 January 2022)).

## 3. Results

### 3.1. Characteristics of the Study Sample

The anthropometric and clinical parameters of the COVID-19 patients are presented in Table 1. According to the guidelines established in the MSD (https://www.msdmanuals.com/professional/SearchResults?query=Normal+laboratory+values (accessed on 6 August 2021)) for normal laboratory values, the COVID-19 patients presented elevated levels of ferritin (463.5 ng/dL) and LDH (276.6 U/L), and a moderate increase in ALT (38 U/L), and AST (38 U/L) enzymes. The principal symptoms were cough (70.3%), dyspnea (47.2%), headache (45.3%), myalgia (40.4%), fatigue (40.4%), and a moderate fever (36.58 ± 0.99 °C), as well as a low oxygen saturation (88.67 ± 10.6).

### 3.2. Allele and Genotype Frequencies

Allele and genotype frequencies of *ABO* polymorphisms in COVID-19 patients and healthy controls are shown in Table 2. Observed and expected frequencies of the six polymorphisms were in Hardy–Weinberg equilibrium. The distribution of the rs651007 *T*/*C*, rs579459 *T/C*, rs495828 *T/G*, and rs8176746 *A*/*C* SNPs were similar in the COVID-19 patients and healthy controls. However, allele and genotype frequencies of the rs8176740 *T/A* (*p* = 0.002 and *p* = 0.0002, respectively) and the rs512770 *T/C* (*p* = 0.003 and *p* = 0.0006, respectively) were different in COVID-19 patients and healthy controls.

### 3.3. Association of ABO rs8176740 T/A and rs512770 T/C SNPs with COVID-19

Under co-dominant, recessive, and additive models, the *TT (Phe*/*Phe)* genotype of the *ABO* rs8176740 *T/A* polymorphism was associated with a high risk of developing COVID-19 (OR = 1.89, 95% CI: 1.23–2.91, pC_Co-dominant_ = 6 × 10^−6^; OR = 1.98, 95% CI: 1.38–2.62, pC_Recessive_ = 1 × 10^−4^; and OR = 1.36, 95% CI: 1.11–1.68, pC_Additive_ = 3 × 10^−3^, respectively). In addition, in the over-dominant model, the *A (Ile)* allele was associated with a low risk of developing COVID-19 (OR = 0.69, 95% CI: 0.51–0.93, pC_Over-dominant_ = 0.014). On the other hand, under co-dominant, recessive, and additive models, the *TT* (Ser/Ser) genotype of the *ABO* rs512770 *T/C* SNP was associated with a high risk of development COVID-19 (OR = 1.87, 95% CI: 1.22–2.88, pC_Co-dominant_ = 2 × 10^−3^; OR = 1.87, 95% CI: 1.30–2.68, pC_Recessive_ = 5 × 10^−4^; and OR = 1.35, 95% CI: 1.10–1.67, pC_Additive_ = 4 × 10^−3^, respectively) (Table 3). All models were adjusted for sex and age.

### 3.4. Linkage Disequilibrium Analysis

The linkage disequilibrium analysis was made separately in those polymorphisms located in the 5′UTR and in those located in the coding region. The rs651007 *T*/*C*, rs579459 *T/C*, and rs495828 *T/G* polymorphisms located on the 5′UTR showed a strong linkage disequilibrium (D’ > 0.95), but none of the haplotypes were associated with a risk of COVID-19. However, the rs8176746 *A*/*C*, rs8176740 *T*/*A*, and rs512770 *T*/*C* polymorphisms located in the coding region showed a moderated linkage disequilibrium (D’ > 0.80). In this block, the *GAC* haplotype was associated with a low risk of developing COVID-19 (OR = 0.53, 95% CI: 0.43–0.66, *pC* = 1 × 10^−6^), whereas the *GTC* and *GAT* haplotypes were associated with a high risk of developing the disease (OR = 5.45, 95% CI: 2.79–10.56, *pC* = 1 × 10^−6^; and OR = 6.33, 95% CI: 2.87–13.9, *pC* = 1 × 10^−6^, respectively) (Table 4).

### 3.5. Association of ABO SNPs with Biochemical Markers in COVID-19 Patients

The relationship of markers associated with damage in patients with COVID-19 [29,30] and the polymorphisms that we detected associated with the disease were analyzed. The markers analyzed were creatinine, ferritin, LDH, CRP, total bilirubin, ALT, AST, hemoglobin, and platelets. In this context, the COVID-19 patients with the rs8176740 *TT* genotype showed a moderate increase in total bilirubin (1.32 ± 2.60 mg/dL, *p* = 0.059) compared with patients with either *AT* (0.81 ± 0.79 mg/dL) or *AA* genotypes (0.86 ± 0.82 mg/dL). In addition, COVID-19 patients with the rs8176740 *TT* genotype had a higher concentration of platelets (295.2 ± 135.2 10^3^/μL, *p* = 0.033) than patients with either *AT* (257.8 ± 106 10^3^/μL) or *AA* genotypes (262.5 ± 113.6 10^3^/μL (Table 5).

## 4. Discussion

Recent studies have shown that the human ABO antibodies play an important role in the process of adhesion of the viral S protein to its ACE-2 receptor. It has been reported that human anti-A antibodies may prevent the attachment between the viral S protein and ACE-2 receptor, so individuals with non-A blood types (O or B blood types) may be less susceptible to SARS-CoV2 infection [14,15,16,17,18]. We studied six polymorphisms located in the ABO gene (rs651007 *T/C*, rs579459 *T/C*, rs8176740 *T/A*, rs8176746 *A/C*, rs495828 *T/G*, and rs512770 *T*/*C*) in patients with COVID-19. Our study found an association of the rs8176740 *T* and rs512770 *T* alleles with a significant susceptibility to developing COVID-19. To the best of our knowledge, this study was the first that described the association of these polymorphisms with the risk of developing of COVID-19. The association of these SNPs and other polymorphisms with COVID-19 in other populations are scarce and controversial. Previously reports that included a genome wide association study (GWAS) showed that some ABO SNPs, such as rs657152 A/C and rs505922 C/T, were associated with a major risk of infection, severity, and mortality in patients with COVID-19 [26,31]. Contrary, recent data have shown that other ABO SNPs were associated with a reduction in SARS-CoV-2 transmission, in terms of the number of infected individuals [10,32,33]. Lehrer and Rheinstein reported that rs505922, rs657152, rs8176746, and rs8176719 ABO SNPs were not associated with COVID-19 in a Caucasian population [27]. As can be seen, the association of the rs512770 *T/C* and rs8176740 *T/A* SNPs with the presence of COVID-19 in different populations are scarce. Nonetheless, data in the literature showed that polymorphisms in the *ABO* sequence can abolish or decrease the enzymatic activity, thus leading to blood group O or weak A/B subgroups [34,35,36]. In addition, previous reports showed that the presence of rs512770 *T*/*C* and rs8176740 *T*/*A* SNPs with more 261delG, as well as nine other SNPs throughout exons 3–7, were involved in blood type O being inactive [34,35,36]. Our data suggested that the blood type O could be involved in an increased risk of developing COVID-19 in our population. We believe future investigations are warranted to understand the contribution of the rs512770 *T* and rs8176740 *T* alleles in the susceptibility to developing COVID-19. On the other hand, we found that the *GTC* and *GAT* haplotypes were associated with a high risk of developing COVID-19, whereas the *GAC* haplotype was associated with a low risk. Interestingly, the haplotype associated with a low risk for COVID-19 did not include the rs8176740 *T* and rs512770 *T* alleles, both of which were associated independently with risk of the disease. This finding corroborated the role of these alleles in the genetic susceptibility to COVID-19, whether they were analyzed independently or as haplotypes.

On the other hand, several studies have shown that infection with the SARS-CoV-2 virus produced frequent multiorgan complications, arterial thromboembolism due to an increase in the biochemical markers of damage such as creatinine, ferritin, LDH, CRP, total bilirubin, ALT, AST, hemoglobin, and high-platelet plasma levels [29,30,37,38]. Additionally, positive and negative associations of the A blood type with the severity of the disease have been reported, and an increase in biochemical biomarkers that led to multiorgan dysfunction [22,39,40]. Our study showed that patients with the rs8176740 *TT* genotype showed a higher concentration of platelets. Nonetheless, the precise mechanism by which the rs8176740 *TT* genotype could be involved in the platelet concentration in COVID-19 patients has not yet been defined. We believe future investigations are warranted to understand the contribution of the *ABO* polymorphism in platelet concentrations in COVID-19.

Finally, the rs8176740 *T*/*A* and rs512770 *T*/*C* polymorphisms with a high risk of developing COVID-19 may also be related to the allelic distribution of these polymorphisms in different populations. In this context, data obtained from the National Center for Biotechnology Information revealed that the individuals from Los Angeles with Mexican ancestry, Mexican mestizos, and Asians had a high frequency of the rs8176740 *T* allele (41%, 46%, and 29%, respectively) compared with Caucasians and Africans (22% and 24%, respectively). On the other hand, the Mexican mestizos, Asians, Africans, and individuals from Los Angeles with Mexican ancestry had a high frequency of the rs512770 *T* allele (45%, 29%, 28%, and 40%, respectively), whereas the Caucasian population had a low frequency of this allele (20%) (https://www.ensembl.org/index.html (accessed on 17 January 2022)).

In summary, this study demonstrated that the rs512770 *T/C*, and rs8176740 *T/A* polymorphisms of the *ABO* gene were associated with an increased risk of developing COVID-19 in a Mexican population. In addition, it was possible to distinguish two haplotypes (*GTC* and *GAT*) associated with an increased risk of developing COVID-19. There was a statistically significant association of the rs8176740 *T*/*A* polymorphism with high-platelet-level plasma. Based on these results, and considering the specific genetic characteristics of the Mexican population, we propose that additional studies in a larger number of individuals and in other populations be undertaken. These studies could help define the true role of these polymorphisms as markers of risk or protection for developing COVID-19.

## Figures and Tables

**Table 1 biomolecules-12-00486-t001:** Demographic and clinical parameters of the COVID-19 patients.

Characteristics	COVID-19 Patients (*n* = 415)
Age (years)	52.2 ± 14.8
Sex n (%)	254 (61) Male
	161 (39) Female
**Clinical symptoms**	
Cough n (%)	292 (70.3)
Dyspnea n (%)	196 (47.2)
Chest Pain n (%)	62 (15)
Headache n (%)	188 (45.3)
Myalgia n (%)	168 (40.4)
Fatigue n (%)	168 (40.4)
Abdominal Pain n (%)	44 (10.6)
Nausea n (%)	58 (14)
Emesis n (%)	40 (9.6)
Diarrhea n (%)	61 (14.6)
Odynophagia n (%)	116 (28)
Fever n (%)	97 (23.3)
Rhinorrhea n (%)	81 (19.5)
Temperature ºC	36.58 ± 0.99
Oxygen saturation (SpO2)	88.67 ± 10.60
Heart rate	87.67 ± 19.32
**Comorbidities**	
Obesity n (%)	238 (57.3)
TDM2 n (%)	124 (29.8)
Hypertension n (%)	142 (34.2)
**Biochemical markers**	
Creatinine (mg/dL)	0.86 (0.64–1.19)
Ferritin (ng/μL)	463.5 (228.5–951)
LDH (U/dL)	276.6 (189–391)
C reactive protein (mg/dL)	13.5 (2.23–89.9)
Total bilirubin (mg/dl)	0.60 (0.44–0.88)
ALT (U/dL)	38 (22.7–68.3)
AST (U/dL)	38 (24–59.8)
Hemoglobin (g/dL)	14.2 (11.8–15.5)
Platelets (10^3^/µL)	248 (198–324.2)

Abbreviations: LDH, lactic acid dehydrogenase; ALT, alanine transaminase; AST, aspartate aminotransferase. Data are expressed as mean ± standard deviation, percentage, and median-percentiles (25–75th).

**Table 2 biomolecules-12-00486-t002:** Allele and genotype distributions of *ABO* gene polymorphisms in COVID-19 patients and healthy controls.

Polymorphic Site		COVID-19 *n* = 415 (n (%))	Controls *n* = 288 (n (%))	* *p*
*ABO* 5′UTR	rs651007 *T/C*			
	Allele			
	*C*	732 (88)	493 (86)	NS
	*T*	98 (12)	83 (14)	
	Genotype			
	*CC*	323 (77.8)	214 (74.3)	
	*CT*	86 (20.7)	65 (22.6)	NS
	*TT*	6 (1.4)	9 (3.1)	
*ABO* 5′UTR	rs579459 *T/C*			
	Allele			
	*T*	727 (88)	492 (85)	NS
	*C*	103 (12)	84 (15)	
	Genotype			
	*TT*	319 (76.9)	213 (74)	
	*TC*	89 (21.4)	66 (22.9)	NS
	*CC*	7 (1.7)	9 (3.1)	
*ABO* 5′UTR	rs495828 *T/G*			
	Allele			
	*G*	727 (88)	492 (85)	NS
	*T*	103 (12)	84 (15)	
	Genotype			
	*GG*	320 (77.1)	214 (74.3)	
	*GT*	87 (21)	64 (22.2)	NS
	*TT*	8 (1.9)	10 (3.5)	
*ABO* Leu266Met	rs8176746 *A/C*			
	Allele			
	*C*	783 (94)	549 (95)	NS
	*A*	47 (6)	27 (5)	
	Genotype			
	*CC*	368 (88.7)	261 (90.6)	
	*CA*	47 (11.3)	27 (9.4)	NS
	*AA*	0 (0)	0 (0)	
*ABO* Phe216Ile	rs8176740 *T/A*			
	Allele			
	*A*	382 (46)	312 (54)	0.002
	*T*	448 (54)	264 (46)	
	Genotype			
	*AA*	101 (24.3)	80 (27.8)	
	*AT*	180 (43.4)	152 (52.8)	0.0002
	*TT*	134 (32.3)	56 (19.4)	
*ABO* Ser74Pro	rs512770 *T/C*			
	Allele			
	*C*	389 (47)	315 (55)	0.003
	*T*	441 (53)	261 (45)	
	Genotype			
	*CC*	101 (24.3)	82 (28.5)	
	*CT*	187 (45.1)	151 (52.4)	0.0006
	*TT*	127 (30.6)	55 (19.1)	

Data are shown as *n* and frequency. * Chi-squared test. NS: Not significant.

**Table 3 biomolecules-12-00486-t003:** Association of *ABO* SNPs with COVID-19.

		*n* (Genotype Frequency)		MAF	Model	OR (95% CI)	pC
*ABO* Phe216Ile	rs8176740 *T/A*						
Control	*AA*	*AT*	*TT*	*T*			
(*n* = 288)	80 (0.278)	151 (0.528)	56 (0.194)	0.46	*Co-dominant*	1.89 (1.23–2.91)	6 × 10^−4^
					*Dominant*	1.19 (0.85–1.68)	0.309
COVID-19	101 (0.243)	180 (0.434)	134(0.323)	0.54	*Recessive*	1.98 (1.38–2.83)	1 × 10^−4^
(*n* = 415)					*Over-dominant*	0.69 (0.51–0.93)	0.014
					*Additive*	1.24 (0.98–1.58)	3 × 10^.3^
*ABO* Pro74Ser	rs512770 *T/C*						
Control	*CC*	*CT*	*TT*	*T*			
(*n* = 288)	82 (0.285)	151 (0.524)	55 (0.191)	0.45	*Co-dominant*	1.87 (1.22–2.88)	2 × 10^−3^
					*Dominant*	1.24 (0.88–1.74)	0.219
COVID-19	101 (0.243)	187 (0.451)	127(0.306)	0.53	*Recessive*	1.87 (1.30–2.68)	5 × 10^−4^
(*n* = 415)					*Over-dominant*	0.74 (0.55–1.01)	0.055
					*Additive*	1.35 (1.10–1.67)	4 × 10^−3^

Abbreviations: COVID-19, severe acute respiratory syndrome coronavirus 2; MAF, minor allele frequency; OR, odds ratio; CI, confidence interval; pC, *p*-corrected. The *p*-values were calculated by the logistic regression analysis, and ORs were adjusted for age and sex.

**Table 4 biomolecules-12-00486-t004:** Frequencies of *ABO* haplotypes in the 5′UTR region and coding region in the study.

Haplotype	5′UTR Region		COVID-19 (*n* = 415)	Controls (*n* = 288)	OR	95% CI	*p*-Value
rs651007	rs579459	rs495828	Hf	Hf			

*C*	*T*	*G*	0.873	0.852	1.19	0.87–1.62	0.261
*T*	*C*	*T*	0.118	0.144	0.79	0.58–1.09	0.179

Haplotype	Coding region		Hf	Hf			
rs8176746	rs8176740	rs512770					

*G*	*T*	*T*	0.452	0.441	1.04	0.84–1.29	0.689
*G*	*A*	*C*	0.332	0.483	0.53	0.43–0.66	1 × 10^−6^
*G*	*T*	*C*	0.088	0.018	5.45	2.79–10.6	1 × 10^−6^
*T*	*A*	*C*	0.049	0.046	1.10	0.66–1.82	0.699
*G*	*A*	*T*	0.072	0.011	6.33	2.87–13.9	1 × 10^−6^

Abbreviations: SARS-CoV-2, severe acute respiratory syndrome coronavirus 2; Hf, haplotype frequency; *p, p*-value. The order of the polymorphisms in the haplotypes is according to the positions in the chromosome 5′UTR region (rs651007, rs579459, rs495828), and coding region (rs8176746, rs8176740, and rs512770).

**Table 5 biomolecules-12-00486-t005:** Association of the *ABO* gene SNPs with biochemical damage markers in the COVID-19 patients.

*ABO*	rs8176740 *T/A* Phe216Ile			*p*-Value
Genotype (*n*)	*AA*	*AT*	*TT*	
Parameters				
Creatinine (mg/dL)	1.08 ± 0.83	1.64 ± 3.14	1.44 ± 1.94	0.292
Ferritin (ng/μL)	682.6 ± 1762.1	885.3 ± 1861.4	788.3 ± 791	0.509
LDH (U/dL)	313 ± 184.3	378 ± 803.8	355 ± 193.1	0.689
C reactive protein (mg/dL)	49.3 ± 78.9	67.2 ± 102.9	73 ± 102.8	0.247
Total bilirubin (mg/dL)	0.86 ± 0.82	0.81 ± 0.79	1.32 ± 2.69	0.059 *
ALT (U/dL)	44.8 ± 35.1	59.10 ± 92.8	54.9 ± 46.1	0.383
AST (U/dL)	42.6 ± 24.8	56.3 ± 92.6	50.8 ± 33.2	0.340
Hemoglobin (g/dL)	13.6 ± 2.74	13.7 ± 2.94	13.4 ± 2.98	0.741
Platelets (10^3^/μL)	262.5 ± 113.6	257.8 ± 106	295.2 ± 135.2	0.033 ^α^
Heart rate	86.4 ± 16.9	88.3 ± 18.7	87.7 ± 20.7	0.765
Temperature °C	36.3 ± 1.37	36.7 ± 0.79	36.6 ± 0.88	0.172
Oxygen saturation (SpO2)	88.4 ± 12.4	89.3 ± 8.96	87.9 ± 11.2	0.576
* **ABO** *	**rs512770 *T/C* Pro74Ser**			***p*-Value**
Genotype (*n*)	*TT*	*TC*	*CC*	
Parameters				
Creatinine (mg/dL)	1.64 ± 2.58	1.52 ± 2.75	1.00 ± 0.65	0.210
Ferritin (ng/μL)	752.7 ± 780.8	861.0 ± 1808.5	752.3 ± 1823.3	0.832
LDH (U/dL)	337.6 ± 184.5	383.6 ± 792	324 ± 236.4	0.687
C reactive protein (mg/dL)	71.7 ± 95.9	66.6 ± 107.6	50.9 ± 76.4	0.366
Total bilirubin (mg/dL)	1.24 ± 2.66	0.88 ± 0.96	0.85 ± 0.81	0.225
ALT (U/dL)	53.6 ± 46.4	56.5 ± 91.6	51.1 ± 37.9	0.866
AST (U/dL)	48.9 ± 34.1	54.9 ± 89.5	47.5 ± 36.5	0.691
Hemoglobin (g/dL)	13.2 ± 3.03	13.8 ± 2.79	13.9 ± 2.85	0.213
Platelets (10^3^/μL)	281.0 ± 127.5	262.8 ± 109.5	272.2 ± 123.2	0.474
Heart rate	86.4 ± 19.9	88.6 ± 20.8	87.4 ± 15.4	0.683
Temperature °C	36.5 ± 1.33	36.7 ± 0.79	36.5 ± 0.81	0.144
Oxygen saturation (SpO_2_)	88.5 ± 9.1	88.1 ± 11.4	89.7 ± 10.7	0.559

Abbreviations: LDH = lactic acid dehydrogenase, ALT = aminotransferase alanine, AST = aminotransferase aspartate. Data for creatinine, ferritin, LDH, CRP, total bilirubin, ALT, AST, hemoglobin, and platelets are expressed as mean ± SD.* Moderate increased total bilirubin (*p* > 0.05). ^α^ High concentration of platelets (*p* < 0.05).

## Data Availability

The data presented in this study are available upon request from the corresponding author.

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
