# Peer review of "The rs8176740 T/A and rs512770 T/C Genetic Variants of the ABO Gene Increased the Risk of COVID-19, as well as the Plasma Concentration Platelets"

_biomolecules, 2022, doi:10.3390/biom12040486_

Round 1
Reviewer 1 Report
The manuscript by Vargas-Alarcón et al entitled “The rs8176740 A/T, and rs512770 T/C genetic variants of the ABO gene increased risk of COVID-19, as well as the plasma concentration platelets.” presents results of association of Covid -19 infection with six ABO gene polymorphisms (SNPs) from 703 clinical samples, using TaqMan genotyping assays. Following six SNPs they claim they genotyped ( rs651007 T/C, rs579459 T/C, rs495928 T/C, rs8176746 A/C, rs8176740 A/T, and 21 rs512770 T/C). The authors demonstrate that the association of rs8176740 A/T, and rs512770 T/C genetic variants increases the risk of Covid-19 infection as well as platelets concentration is plasma. Others 4 SNPs are not associated with Covid-19 infection
The manuscript qualifies publication but the authors need to correct and ensure that out of the 6 SNPs analyzed rs495928 T/C is incorrect ID. I assume this to be a typographical error as rs495828 is the correct ID. The author need to ensure the investigated the correct SNP, and correct it throughout the manuscript.
Author Response
Thank you very much for your comments and suggestions to Manuscript ID biomolecules-1624597
We would like to thank the Reviewer for their comments; they have helped to improve the manuscript
Comments to the Author:
Reviewer #1: Comments and Suggestions for Authors
The manuscript by Vargas-Alarcón et al entitled “The rs8176740 A/T, and rs512770 T/C genetic variants of the ABO gene increased risk of COVID-19, as well as the plasma concentration platelets.” presents results of association of Covid -19 infection with six ABO gene polymorphisms (SNPs) from 703 clinical samples, using TaqMan genotyping assays. Following six SNPs they claim they genotyped (rs651007 T/C, rs579459 T/C, rs495928 T/C, rs8176746 A/C, rs8176740 A/T, and 21 rs512770 T/C). The authors demonstrate that the association of rs8176740 A/T, and rs512770 T/C genetic variants increases the risk of Covid-19 infection as well as platelets concentration is plasma. Others 4 SNPs are not associated with Covid-19 infection
The manuscript qualifies publication but the authors need to correct and ensure that out of the 6 SNPs analyzed rs495928 T/C is incorrect ID. I assume this to be a typographical error as rs495828 is the correct ID. The author need to ensure the investigated the correct SNP, and correct it throughout the manuscript.
Answer: Agree with the reviewer’s comment, we apologize for this typographical error. In order to clarified this point, we re-verified the ID of the SNPs studied in this work, and corrected these errors. In this context, we changed the ID incorrect (rs495928 T/C) by ID correct (rs495828 T/G) along the manuscript.
Reviewer 2 Report
The manuscript presents analysis of SNPs in ABO (blood group) gene on succeptibility to SARS-CoV-2 infection. The authors found an association with rs8176740 and rs512770 SNPs that encode amino acid substitutions, as well as association of the former with platelet count and billirubin level.
The study is well designed and described. I have almost no comments except the following:
- please indicate a period of patient recruitment as it can tell which SARS-CoV-2 types were circulating at that time.
- I would also try to analyze association of SNPs (data of table 5) not only with absolute levels of the clinical parameters but also with percentages of abnormal values in each group (as for some of them the spread of values is really high).
Author Response
Thank you very much for your comments and suggestions to Manuscript ID biomolecules-1624597
We would like to thank the Reviewer for their comments; they have helped to improve the manuscript
Comments to the Author:
Reviewer #2: Comments and Suggestions for Authors
The manuscript presents analysis of SNPs in ABO (blood group) gene on susceptibility to SARS-CoV-2 infection. The authors found an association with rs8176740 and rs512770 SNPs that encode amino acid substitutions, as well as association of the former with platelet count and bilirubin level.
The study is well designed and described. I have almost no comments except the following:
1.- Please indicate a period of patient recruitment as it can tell which SARS-CoV-2 types were circulating at that time.
Answer: According with the Reviewer’s comment, we added period of recruitment of the patients with COVID-19. In this context, we modified the following phrase “The patients were recluted from different Hospital centers, 162 from the Instituto Nacional de Rehabilitación “Luis Guillermo Ibarra Ibarra”, 108 from the Hospital Juárez de México, and 145 from the Instituto Nacional de Cardiología Ignacio Chávez.” By “The patients were recruited from April 2020 to January 2021 of different Hospital centers, 162 from the Instituto Nacional de Rehabilitación “Luis Guillermo Ibarra Ibarra”, 108 from the Hospital Juárez de México, and 145 from the Instituto Nacional de Cardiología Ignacio Chávez.” in material and methods section.
2.- I would also try to analyze association of SNPs (data of table 5) not only with absolute levels of the clinical parameters but also with percentages of abnormal values in each group (as for some of them the spread of values is really high).
Answer: We appreciate your suggestion; however, we think that adding (in addition to the absolute levels of the clinical parameters in Table 5 or an extra table) the percentage of abnormal values of the clinical parameters might be repetitive and confusing for readers. In this context, we decided not to add information related to percentages of abnormal values for each group in Table 5, but we could do that if the reviewer considered this necessary.